# Photodynamic Therapy Combined with Antibiotics or Antifungals against Microorganisms That Cause Skin and Soft Tissue Infections: A Planktonic and Biofilm Approach to Overcome Resistances

**DOI:** 10.3390/ph14070603

**Published:** 2021-06-23

**Authors:** Vanesa Pérez-Laguna, Isabel García-Luque, Sofía Ballesta, Antonio Rezusta, Yolanda Gilaberte

**Affiliations:** 1Max Planck Institute for Evolutionary Biology, 24306 Plön, Germany; 2Department of Microbiology, School of Medicine, University of Sevilla and Instituto de Biomedicina de Sevilla (IBIS), 41009 Sevilla, Spain; igarcial@us.es (I.G.-L.); mudarra@us.es (S.B.); 3Aragon Health Research Institute (IIS Aragón), 50009 Zaragoza, Spain; arezusta@unizar.es (A.R.); ygilaberte@salud.aragon.es (Y.G.); 4Department of Microbiology, Miguel Servet University Hospital, 50009 Zaragoza, Spain; 5Department of Microbiology, Preventive Medicine and Public Health, University of Zaragoza, 50001 Zaragoza, Spain; 6Department of Dermatology, Miguel Servet University Hospital, 50009 Zaragoza, Spain

**Keywords:** photoinactivation, resistance, antimicrobial agents, SSTI, MDR

## Abstract

The present review covers combination approaches of antimicrobial photodynamic therapy (aPDT) plus antibiotics or antifungals to attack bacteria and fungi in vitro (both planktonic and biofilm forms) focused on those microorganisms that cause infections in skin and soft tissues. The combination can prevent failure in the fight against these microorganisms: antimicrobial drugs can increase the susceptibility of microorganisms to aPDT and prevent the possibility of regrowth of those that were not inactivated during the irradiation; meanwhile, aPDT is effective regardless of the resistance pattern of the strain and their use does not contribute to the selection of antimicrobial resistance. Additive or synergistic antimicrobial effects in vitro are evaluated and the best combinations are presented. The use of combined treatment of aPDT with antimicrobials could help overcome the difficulty of fighting high level of resistance microorganisms and, as it is a multi-target approach, it could make the selection of resistant microorganisms more difficult.

## 1. Introduction

### 1.1. The Problem of Skin and Soft Tissue Infections

Skin and soft tissue infections (SSTIs) are defined as clinical entities of variable presentation, etiology, and severity that involve microbial invasion of the layers of the skin and underlying soft tissues. Related to clinical manifestations of the wide range of pathologies they represent, the minimum diagnostic criteria are erythema, edema, warmth, and pain or tenderness. The affected area may also become dysfunctional depending on the severity of the infection, and, much more relevantly, patient comorbidity can easily transform a normally mild infection into a rapidly advancing threat to life [1,2,3]. Complicated forms of SSTI (cSSTI) may need, apart from antibiotic or antifungal treatment, surgical procedures, or have significant underlying co-morbidities such as diabetes, systemic immunosuppression, or neurological diseases [4,5,6].

Their treatment has become more challenging because of the increase in the frequency and severity of infections mainly due to the ageing of the general population, the increased number of critical and immunocompromised patients, and because of the emergence of resistance to many of the antimicrobial agents commonly used to treat SSTIs in the past [7,8,9].

### 1.2. Antimicrobial Resistance in Skin and Soft Tissue Infections Causal Agents

Antimicrobial resistance (AMR) makes treatments more tedious and it has adverse consequences such as prolonged hospitalization, increased medical expenses, overburdened public health system, and even increased mortality rates [10].

The main causative agent of SSTI is *Staphylococcus aureus*, which is one of the bacteria most frequently involved in the problem of AMR. Other causal agents of a high percentage of SSTI, also involved in AMR problem, are *Enterobacteriaceae* and *Pseudomonas aeruginosa* (e.g., 45.9% of hospital-acquired SSTI in North America are caused by *S. aureus*, 10.8% by *P. aeruginosa*, and 8.2% by *Enterococcus* spp.) [1]. In fact, in 2017 the World Health Organization (WHO) published the first list of bacteria for which new antibiotics were urgently needed [11] (Table 1). Carbapenem-resistant or third generation cephalosporin-resistant *Enterobacteriaceae* (e.g., *Escherichia coli*) and carbapenem-resistant *P. aeruginosa* are classified in this list within the most dangerous class (Priority 1: critical) and methicillin-resistant or vancomycin intermediate and resistant *S. aureus* is classified as “Priority 2: high”.

Carbapenem-resistant *Acinetobacter baumannii* completes the list of bacteria classified as “Priority 1”. *A. baumannii*-associated SSTI is an emerging infection in patients who experience trauma; although it causes few cases, usually they are severe [12].

Regarding atypical mycobacteria, which also cause SSTI on certain occasions, they are highly resistant to antibacterial treatments due to the special characteristics of their wall (the mycobacterial cell has four main layers: (i) the plasma membrane, (ii) the peptidoglycan–arabinogalactan complex, (iii) an asymmetrical outer membrane covalently linked with mycolic acids, and (iv) an outermost capsule). They are a globally established priority for which innovative new treatments are urgently needed according to the WHO [11].

On the other hand, the recent emergence of fungi that are resistant to more than one class of antifungal drug is a serious concern, especially because currently only three primary classes of agents are used to treat invasive fungal infections: (1) Azoles (Fluconazole, voriconazole, posaconazole), (2) Echinocandins (Caspofungin, micafungin, anidulafungin) and (3) Polyne (Amphotericin B). *Candida glabrata*, *Candida krusei* and *Candida auris* are species with intrinsic or high rates of resistance against the first, *Cryptococcus* spp. and *Fusarium* spp. against the second, and *Candida auris* and *Aspergillus terreus* against the last [13].

With regard to yeasts that cause SSTI, *Candida* spp. stand out as causative agents. They are implicated in the AMR problem due to the increasingly frequent existence of fluconazole resistant strains [13,14].

Finally, dermatophytes and non-dermatophytes molds cause cutaneous and subcutaneous infections than are often chronic and recalcitrant. Invasive infections are rare but occur especially in immunocompromised and debilitated individuals. Its treatment is a great challenge especially in these patients. [15,16].

### 1.3. Antimicrobial Photodynamic Therapy Combined with Antibiotics or Antifungals to Treat SSTIs

Antimicrobial photodynamic therapy (aPDT) is based on the use of photosensitizer molecules (PS) that are activated by light in the presence of oxygen. Reactive oxygen species (ROS) are generated in the photodynamic reaction resulting in a toxic effect for microorganisms that is capable of destroying them (Figure 1). Hence, aPDT has been proposed as an alternative treatment for SSTIs [17,18].

Among the advantages of the aPDT is the photoinactivation of the microorganisms independently of their pattern of AMR, without its use contributing to the selection of drug-resistant strains, and its broad spectrum of activity; therefore it does not require a precise microbial diagnosis and it is very useful for infections caused by several microorganisms [19,20]. By contrast, the most import limitation is the possibility of microbial regrowth by those who have not been inactivated during irradiation [17,18,21]. The combination of aPDT with antibiotic or antifungal treatment is a promising approach to the fight against infectious diseases due to aPDT’s ability to increase the susceptibility of microorganisms to antimicrobial drugs and minimize the possibility of the regrowth after aPDT [22,23,24].

### 1.4. Objective

We present a review of articles related to in vitro activities of aPDT in combination with antibiotics or antifungals, focusing on microorganisms that cause SSTIs. The aim is to combine the information that exists about each microorganism (gather the bases of knowledge). This review serves as a starting point for new combined treatment research (expand and apply knowledge).

The questions that are intended to be answered are: (1) Which microorganisms that cause SSTIs have been exposed in vitro to combinations of aPDT and antimicrobials? (2) Which methodologies have been used? (3) Are there an additive or synergistic antimicrobial effect? (4) Which are the best combinations clinically and microbiologically?

## 2. Methodology

### 2.1. Eligibility Criteria

We have considered in vitro studies that used antimicrobial agents (antibiotics or antifungals) plus aPDT against microorganisms that cause SSTIs. The specific requirements for inclusion of the studies were (1) in vitro studies in planktonic state or forming biofilm and ex vivo studies on the skin or mucous membrane; (2) aimed to inactivate identified bacteria, or fungi that cause SSTIs; (3) used antibiotics or antifungals as a fundamental part of the treatment; (4) used aPDT as a fundamental part of the treatment; and (5) published in indexed journals and written in English or Spanish.

### 2.2. Study Selection, Data Collection Process, and Characteristics

The keywords used for the search in Pubmed and Embase library databases were: photodynamic therapy, PDT, antimicrobial photodynamic therapy, aPDT, photodynamic antimicrobial chemotherapy, PACT, photoinactivation, photodynamic inactivation, PDI, combination, combined treatment, antimicrobial agents, antibiotics, and antifungals. The last search was carried out on 1 June 2021 and no time limits in the past were applied. Research papers that meet the eligibility criteria (Section 2.1) were included. A huge number of studies contained the keywords; nevertheless, after applying the eligibility criteria, the number was drastically reduced to a total of 33.

The data recapitulated were: (1) etiological agent of SSTIs; (2) type of study: in vitro (planktonic or forming biofilm) and ex vivo; (3) antimicrobial methodology: antibiotics or antifungals used and their application and concentration; (4) aPDT methodology: PS used and parameters of irradiation (source type, wavelength, intensity, and fluence); (5) effect of combined treatment against microorganism.

The 33 included studies were grouped depending on the etiological agent of SSTIs. The structure used is as follows:

-Gram-positive bacteria: *Staphylococcus* spp. (Section 3.1: Section 3.1.1. S. aureus and Section 3.1.2. *Staphylococcus epidermidis* and *Staphylococcus hemolytic)*

-Mycobacteria: *Mycobacterium fortuitum* (Section 3.2.)

-Gram-negative bacteria: *E. coli, P. aeruginosa*, and *A. baumannii* (Section 3.3, Section 3.4, Section 3.5 respectively)

-Fungal infectious agents: Yeast (Section 3.6. *Candida* spp.), dermatophytes, and molds (Section 3.7).

## 3. Results of Studies on In Vitro aPDT Combined with Antimicrobial Agents against Infectious Microorganism of Skin and Soft Tissues

### 3.1. Staphylococcus *spp.*

#### 3.1.1. Staphylococcus aureus

*S. aureus*, frequently involved in the problem of AMR, is the main causative agent of SSTI and represents one of the most important pathogens involved in cSSTI [2,25,26]. For this reason, this is the bacteria with more studies trying the combination of different antibiotics and PS published so far. Table 5 summarizes the studies concerning in vitro aPDT plus other antimicrobial treatments against *S. aureus*.

##### Porphycene and Porphyrin Studies

To our knowledge, one of the most recent articles about aPDT and antimicrobials’ combination was published by Nieves et al. They report the synthesis of a new porphycene namely 2-aminothiazolo[4,5-c]-2,7,12,17-tetrakis(methoxyethyl)porphycene (ATAZTMPo)-gentamicin conjugate. It outperforms the antimicrobial effect against *S. aureus* and *E. coli* of the two components delivered separately. This novel photoantimicrobial agent may be used to enhance the therapeutic index of gentamicin, broaden the spectrum of pathogens against which it is effective, and reduce its side effects [27].

The combination of gentamicin with porphyrins also seems to be effective against *S. aureus* [26,28].

5-aminolevulinic acid (5-ALA), a pro-drug that, once metabolized by proliferating bacteria, is converted into the natural PS protoporphyrin IX (PpIX), combined with a low concentration of gentamycin (2 μg/mL) improved qualitative and quantitative antibacterial effects against *S. aureus* biofilms. The proposed explanation was that photoactivation generated ROS which damages or kills the cells, while gentamicin, even at low doses, completes the eradication. The mechanism of action of gentamicin is based on its capacity to bind tightly to the 30S subunit of the bacterial ribosome impairing protein synthesis and the bacterial cells die. The fundamental requirement for gentamicin to act as an antibiotic is the need to penetrate the membrane of the bacterial cell, and aPDT can possibly damage the membrane thus increasing antibiotic penetration [28].

Deuteroporphyrin-aPDT combined with different antibiotics included gentamicin, vancomycin, rifampicin, fusidic acid, and oxacillin did not detect synergistic effect except for oxacillin. They concluded that only the combination of deuteroporphyrin-aPDT plus oxacillin had potential for aPDT to improve traditional antibiotic treatment with cell wall synthesis inhibitors [29].

Regarding the combination of porphyrins with other antibiotics different from gentamycin, when ALA-aPDT was combined with netilmicin, vancomycin or cefaclor against *S. aureus* biofilms, synergistic bactericidal effect was observed. Destruction occurred predominantly in the upper layer of the biofilm, and in a strain-dependent manner. That is why they suggested that a drug sensitivity test should be performed in advance [30].

Meso-tetrakis(*N*-methyl-4-pyridyl) porphine tetra-tosylate (*TMP*), a cationic porphyrin combined with vancomycin, was highly effective against *S. aureus* biofilms. It seems that the combination blocks cell wall synthesis, and the damaged biofilms may be more susceptible to host defenses which could be useful for biofilms adhering to medical implant surfaces [31].

In contrast, the combination of other cationic porphyrin, 5,10,15,20-tetrakis[4-(3-*N*,*N*-dimethylaminopropoxy) phenyl] porphyrin (TAPP)-aPDT with the cell wall inhibiting antibiotics ceftriaxone or vancomycin did not significantly increase the aPDT effect on planktonic *S. aureus*. However, the bactericidal effect of TAPP-aPDT was additive with the protein synthesis inhibitors chloramphenicol and tobramycin against *S. aureus* and *E. coli*, and was synergistic against methicillin resistant *S. aureus* (MRSA) and *S. epidermidis*. It seems that vancomycin and ceftriaxone presented limited effects when combined with TAPP-aPDT; nevertheless, tobramycin and chloramphenicol reached additive effects for *S. aureus* and *E. coli* and synergy against MRSA and *S. epidermidis*, even when sub-minimum inhibitory concentration (MIC) levels of TAPP and antibiotics were used. The results suggest that even sub-MIC levels of photo-activated TAPP could be used to boost the activity of waning antibiotics [32].

Tetracationic porphyrin *meso*-tetrakis(1-methylpyridinium)porphyrin (Tetra-Py^+^-Me)-PDT combined with ampicillin significantly reduced *S aureus* survival in both states (planktonic and biofilm) [33].

Finally, the endogenous porphyrins accumulated by some microorganism, such as *E. coli* or *S. aureus*, combined with different wavelengths of light (blue, 470 nm, or red, 625 nm) in the presence of ciprofloxacin were more effective than the antibiotic alone [34].

Overall, there are multiple promising combinations of antibiotics and porphyrins-PDT which could increase the efficacy in vitro of conventional antibiotics against *S. aureus* (See Table 5).

##### Phenothiazine Studies

Another chemical group of PS for PDT are phenothiazinium dyes, such as methylene blue (MB) and toluidine blue O (TBO).

MB-aPDT combined with chloramphenicol showed an additive effect against planktonic *S. aureus* [32], while ciprofloxacin plus MB-aPDT was synergistic either in planktonic and in biofilm forms [35]. As usual, bacteria forming biofilms were less affected than bacteria in planktonic phase and a bigger effect was seen when ciprofloxacin was applied after aPDT rather than before or simultaneously. Kashef et al. using MB or TBO-aPDT with linezolid against biofilm resulted in a greater reduction in the viable count of bacteria than either of them separately. TBO-aPDT plus linezolid gave the greatest reduction (although the killing was only <3 log_10_ reduction in viable count for any of the strains) [36].

TBO-aPDT combined with gentamicin and incubated 45-min prior to irradiation showed positive effects against *S. aureus* strains in planktonic state, both in a reference strain and in a multidrug resistant (MDR) clinical isolate; they showed that TBO-aPDT treatment permeabilized the bacterial membranes, promoted gentamicin cellular accumulation and the intracellular ROS generation by the combination was much higher than that of single treatment groups [37].

Our group explored the effect of MB-aPDT, alone or in combination with gentamicin, against planktonic and biofilm *S. aureus*. The addition of gentamicin (concentrations that by themselves do not achieve any effect) caused the complete photoelimination in the case of planktonic *S. aureus*, with a lower MB-PDT dose, whereas it did not produce any change in biofilm [38]. In a previous study, it was proven that concentrations of linezolid or mupirocin which did not harm the bacteria also exert a synergistic effect when they were combined with MB, also reducing the concentration of the PS needed to photoinactivate planktonic *S. aureus* [39]. However, recently we have verified in a superficial abrasion mouse model of *S. aureus* skin infection that the addition of mupirocin to MB-aPDT treatment improved antimicrobial activity but it did not enhance wound healing [40].

##### Xanthene Studies

Rose Bengal (RB, a xanthene dye) has been rehearsed combined with mupirocin, linezolid, or gentamicin with a synergistic bactericidal effect in planktonic *S. aureus* [39]. Additionally, the combination of RB-aPDT plus gentamicin was evaluated against *S. aureus* biofilms. However, only high concentration of RB (64 μg/mL) and gentamicin (40 μg/mL) showed a synergistic effect against biofilms [41]. It is interesting that the combination of RB-aPDT with methicillin significantly reduced the MIC of methicillin either of MRSA or methicillin sensitive *S. aureus* [26].

#### 3.1.2. *Staphylococcus epidermidis* and *Staphylococcus haemolyticus*

*S. epidermidis* and *S. haemolyticus* are part of the skin flora of humans but in specific circumstances, such as with immunocompromised or hospitalized patients with catheters or other surgical implants, they are well-known opportunistic pathogens which cause local or systemic infections. The highly antibiotic-resistant phenotype and their ability to form biofilms makes infections difficult to treat [42,43,44,45].

The study of Barra et al. has explored 5-ALA-aPDT plus gentamicin at a low concentration against clinical isolates of *S. haemolyticus*, *S. epidermidis*, and *S. aureus* biofilms; they reported a synergistic effect being *S. haemolyticus* the most sensitive to photoinactivation [28].

Other porphyrins, such as TMP-aPDT combined with vancomycin in biofilms [46] or TAPP-aPDT combined with chloramphenicol or tobramycin in planktonic cells, showed a higher efficacy against *S. epidermidis* than any of them separately. The authors hypothesized that TAPP could act in combination with lower concentrations of antibiotics providing a controlled release of the antibiotic, and in consequence improving their efficacy to maintain an antimicrobial effect [32].

In conclusion, multiple antibiotics in combination with different types of PDT based on protoporphyrins showed improved bactericidal effects against planktonic *S. epidermidis* or forming biofilms [28,32,46].

Table 5 includes a summary of all the studies of aPDT combined with antibiotics. Table 2 summarizes in detail the methodology and the results of different studies that reported the in vitro activity of aPDT plus antibiotics against *S. haemolyticus* and *S. epidermidis* (Table 2: Biofilm Studies and biofilm state, planktonic—Table 2: Planktonic Cell Studies).

### 3.2. Mycobacterium fortuitum

*M. fortuitum* is an atypical mycobacteria involved in skin infections very difficult to treat, thus usually causes refractory infections [47,48].

The combination of MB-aPDT with ciprofloxacin, moxifloxacin, or amikacin against a clinical isolate had in vitro synergistic antimicrobial effect (the latter also was tested in vivo to treat a model of keratitis in rabbits resulting in significantly less bacterial burden). Sublethal concentrations of antibiotic plus sublethal dosage of MB reached a mycobacterial survival reduction in the colony forming units (CFU) of at least 2 log_10_ lesser compared to the reduction achieved with antibiotics alone, although this effect became insignificant when higher doses of antibiotics were used. The maximum degree of inactivation was achieved by the combination of MB-aPDT with ciprofloxacin or moxifloxacin (≥4 log_10_ reduction, with amikacin (~3 log_10_ reduction) [49]. Table 5 and Table 3 summarize the methodology and the results of this study.

### 3.3. Escherichia coli

*E. coli* is the best-known bacteria of the *Enterobacteriaceae* family [50]. Even though some serotypes of *E. coli* can cause serious disease, most strains are harmless and can only act as opportunistic pathogens [51].

Table 5 includes a summary of the in vitro aPDT plus antibiotics against *E. coli* studies and Table 4 summarizes the methodology and the results of Gram-negatives studies.

Dastgheyb et al., Ronqui et al., and Nieves et al. investigated not only strains of *E. coli* but also *S. aureus.* As has been previously commented, results are similar in both species [27,32,35].

#### 3.3.1. Porphycene Study

Nieves et al. report the synthesis of a new porphycene, ATAZTMPo -gentamicin conjugate, that is able to reduce 8 log_10_ of surviving planktonic *E. coli* while it did not exert any effect in darkness. The gentamicin concentration used was close to the reported MIC (3.8 μg/mL and 4 μg/mL respectively). Nevertheless, MIC values were determined at inoculum sizes 100–1000-fold, more diluted than those used in this study, and the antimicrobial effect usually was evaluated after 24 h of incubation with the antibiotic, while only 1 h of preirradiation incubation was allowed in these experiments; therefore, it is not surprising the lack of gentamicin toxicity in darkness [27].

#### 3.3.2. Phenothiazine and Porphyrin Studies

Among Dastgheyb et al. and Ronqui et al. studies, the best results against planktonic *E. coli* were obtained by the later using MB-aPDT followed by ciprofloxacin. The former evaluated TAPP-aPDT associated with tobramycin or chloramphenicol achieving only additive antibacterial effect [32]. The later achieved a reduction of approximately 7 logs of *E. coli* in planktonic and biofilm using MB-aPDT followed by ciprofloxacin (while the reduction against *S. aureus* biofilms was 5.4 logs). The most remarkable is that the synergistic effect of aPDT plus ciprofloxacin overcame the resistance of biofilm to aPDT. On the other hand, when the antibiotic is applied after aPDT, lower concentrations than the MIC of ciprofloxacin could be used, since the first sub-MIC led to bacterial reduction of both *S. aureus* and *E. coli* in planktonic state. They hypothesize that this combination achieves a higher effect in biofilm than in planktonic, and higher in *E. coli* than in *S. aureus*, because MB-aPDT works worse on *E. coli* biofilm and the effect of the addition of antibiotic therapy may be more evident [35]. Pereira et al. studied the effect of ciprofloxacin and norfloxacin against planktonic *E. coli*, including MDR clinical isolates, when the strains were irradiated, and they observed because of the endogenous porphyrins an increase in the inhibition halo diameter, indicating that these combinations were synergistic [34]. Other combinations, such as ampicillin or chloramphenicol with aPDT using 5,10,15,20-tetrakis(1-methylpyridinium-4-yl) porphyrin tetra-iodide, achieved a greater increase in MDR *E. coli* killing, especially with ampicillin, at sub-inhibitory and inhibitory concentrations [52]. On the other hand, Costa et al. in a comparative study of the effect of MB-aPDT alone or in combination with ceftriaxone against planktonic clinical isolates of Gram-negative bacteria (*E. coli*, *Klebsiella aerogenes*—formerly *Enterobacter aerogenes*—and *Klebsiella pneumoniae* resistant to third-generation cephalosporins) concluded that the combination did not cause an effect on bacterial viability greater than that MB-PDT itself [53].

#### 3.3.3. Chlorophyll Study

The combination of ciprofloxacin, amikacin, or colistin with aPDT using chlorin e6 improves the antibacterial activity of antibiotics against planktonic *E. coli*, being the best combination with ciprofloxacin [51].

### 3.4. Pseudomonas aeruginosa

*P. aeruginosa* is an opportunistic human pathogen especially causing infections in chronic ulcers and burns. Table 5 includes a summary of the in vitro aPDT plus antibiotics against *P. aeruginosa* studies and more information on the different methodologies can be consulted in Table 4.

#### 3.4.1. Porphyrin Studies

Sequential treatments of TMP-aPDT followed by tobramycin against *P. aeruginosa* biofilms showed a synergistic effect. The survival bacteria decreased and biofilms pretreated with TMP-aPDT were substantially more sensitive to tobramycin than untreated biofilms [54].

Different studies employed endogenous porphyrins as the PS. The combination of green light (532 nm) and gentamicin resulted in an antibacterial effect against planktonic *P. aeruginosa* [55]. On the other hand, the combination of sublethal doses of blue light (410 nm) and antibiotics (gentamicin, meropenem, or ceftazidime) reduced the MIC against some planktonic *P. aeruginosa* strains that displayed MDR and extensive drug resistance profiles [56].

#### 3.4.2. Phenothiazine Studies

Another study evaluated the in vitro phototoxicity of MB plus ofloxacin against sensitive or resistant *P. aeruginosa* strains that tolerated ofloxacin. MB-aPDT in combination with antibiotic significantly reduced the viability of *P. aeruginosa* strains compared to either one in monotherapy [57]. Recently, our group showed how gentamicin enhances MB-aPDT-induced bacterial photinactivation of planktonic and biofilm *P. aeruginosa*; an in vitro synergistic effect against both states of *P. aeruginosa* was found, therefore the authors hypothesized that this combination could be useful to manage difficult-to-treat SSTI caused by *P. aeruginosa* [38].

#### 3.4.3. Xanthene Studies

The use of RB-aPDT combined with CAMEL or pexiganan (antimicrobial peptides) achieves a total elimination of different *P. aeruginosa* clinical isolates, including MDR and XDR strains, in contrast to none or partial reduction when the treatments are applied separately. In addition, they demonstrated that the combination is safe, without harmful effects in human keratinocytes [58]. On other hand, Ilizirov et al. studied the combination of RB-aPDT with sulfanilamide against sensitive and resistant *P. aeruginosa* clinical isolates reporting the absence of additional effect. They hypothesize that it could be because of the very low rates of bacterial envelope damage by RB at sub-MIC, which was not sufficient for enhancing the action of sulfanilamide and they suggest that maybe is better to choose antibiotics which affect the same cell target sites in order to achieve amplification of their activity [26].

### 3.5. Acinetobacter baumannii

*A. baumannii* is a threatening human pathogen with outstanding capability to acquire AMR [11] although it rarely causes skin infection SSTIs.

Different antibiotics were combined with RB-aPDT or only blue light to excite endogenous PS against extremely drug resistant *A. baumannii* clinical isolates. The final result was the effective eradication. The bactericidal effect of antibiotics was enhanced with sublethal aPDT or blue light addition applied after them. Moreover, they measured the production of ROS and claimed that its increase with the combined treatment could explain the synergistic activity observed. Table 5 includes a summary of this study (and more information can be seen in Table 3) that stands out for covering all antibiotic categories as well as all antimicrobial mechanisms of action and the wide variety of methods used to test synergy (diffusion assays—disk diffusion and E-test, serial dilution methodology—checkerboard assay, and CFU counting- and time kill curves method—postantibiotic effect) [59].

### 3.6. Candida *spp.*

Candida spp. cause skin and oral and genital mucosa infections and *C. albicans* is the most relevant one [60].

Table 5 includes a summary of the different combinations studied using in vitro aPDT plus antifungals against yeasts that cause SSTIs; specific details of the methodology of these studies are provided in Table 6.

The effect of MB-aPDT plus fluconazole was evaluated against fluconazole-resistant *C. albicans*, *Candida glabrata*, and *Candida krusei*. A synergistic combination effect against the strains of *C. albicans* and *C. glabrata*, but not against *C. krusei*, was found. The effect of MB-aPDT alone against *C. krusei* was not statistically significant compared to the effect of its combination with fluconazole [61].

Snell et al. showed that fluconazole did not enhance *C. albicans* killing induced by aPDT using TMP. However, miconazole improved the fungicidal activity of aPDT using either TMP or MB [62].

### 3.7. Dermatophytes and Moulds

Table 5 includes a summary of the combination treatment studies using in vitro aPDT plus other compounds against dermatophytes and molds that cause SSTIs and an extended version can be consulted in Table 6.

*Trichophyton rubrum* causes of athlete’s foot, fungal infection of nail and ringworm, worldwide athlete’s foot, onychomycosis, jock itch, and ringworm. Morton et al. reported that clotrimazole combined with RB-aPDT had the potential to reduce the MIC of the antifungal drug against spores of *T. rubrum* clinical isolate. This occurred when pre-treatment with the antifungal (sublethal dose) was followed by RB-aPDT. When the order of combination was changed, no reduction in the antifungal MIC was observed [63].

*Fonsecaea monophora* and *Fonsecaea pedrosoi* are the main causative agents of chromoblastomycosis in Southern China. It is a chronic skin and subcutaneous fungal infection with low cure and high relapse rates. Hu et al. (2015) treated five refractory and complex cases of chromoblastomycosis with 5-ALA-aPDT combined with oral antifungal drugs. The isolates were evaluated for susceptibility to terbinafine, itraconazole, and voriconazole and 5-ALA-aPDT in vitro, revealing sensitivity to the antifungals, with 5-ALA-aPDT altering the cell wall and increasing ROS production. The results showed that there was an unclear synergistic effect of itraconazole plus 5-ALA-aPDT [64]. Altogether, they conclude that 5-ALA-aPDT combined with oral antifungal drugs is a promising method for the treatment of refractory and complex cases of chromoblastomycosis. This idea was also previously defended by the same authors in a previous clinical case of chromoblastomycosis caused by *F. monophora* that was successfully treated with terbinafine plus 5-ALA-aPDT. This study was also supported by in vitro experiments where they showed the good outcome of 5-ALA-aPDT applied for the inhibition of *F. monophora* [65].

*Exophiala* spp. is an ubiquitous fungal species commonly found in soil and plants which causes chromoblastomycosis [66]. On the other hand, *Fusarium solani* and *Fusarium oxysporum* are responsible for approximately 60% and 20% of the cases of fusariosis, respectively, which is the second most common mold infection in humans after aspergillosis [67]. Gao et al. investigated both planktonic suspensions and biofilms of *E. dermatitidis* and *Fusarium* spp. with MB-aPDT combined with standard antifungal treatments (itraconazole, voriconazole, posaconazole, and amphotericin) achieving encouraging results. The pretreatment with MB-aPDT made them more susceptible to antifungals either in planktonic cultures or in biofilms. Therefore, the combination may help to enhance the antifungal susceptibility to overcome problems with drug resistance issues, and has the potential to reduce antifungal drug dosages decreasing their toxicity [15]. According to the authors, this may be due to an increased membrane permeability caused by aPDT, as suggested previously for *C. albicans* [68].

*Sporothrix globosa* is an etiological agent of sporotrichosis whose most common clinical manifestation is cutaneous and subcutaneous nodular lesions with lymphangitis involvement. Currently, AMR and complications are the major concerns, especially in patients who have liver disorders, children, and pregnant women. Li et al. compared the efficacy in the inactivation of *S. globosa* of MB-aPDT alone or combined with itraconazole in planktonic culture and in a murine model. The combined treatment offered better results in terms of inactivation percentage and improvement of the lesion size. Therefore, they conclude that MB-aPDT could be an effective adjuvant therapy for resistant infections caused by *Sporothrix* spp. [69]. In fact, our group treated a patient with cutaneous sporotrichosis using intralesional 1% MB-aPDT in combination with intermittent low doses of itraconazole obtained complete microbiological and clinical response [70].

## 4. Summary of Evidence and Limitations

In the main, the combination of antibiotics or antifungals with aPDT against in vitro SSTI-etiological agents seems to be beneficial. Combined therapy is more effective than individual treatments alone and often the effects are greater than additivity, i.e., there is synergy. Among the effects reported, the following stand out: (i) the increase in percentage of microbial inactivation; or (ii) the same inactivation percentage is achieved using lower doses of antimicrobials.

It is remarkable that in some cases, drug sensitivity of resistant strains can be restored by combining antibiotics/antifungals with aPDT [15,26,59].

The highlight combinations and the best treatment protocols supported by the existing evidence of in vitro studies on combined aPDT therapies against SSTI-causing agents are included in Table 7. Nevertheless, there is not enough evidence to establish the best combination against each causal agent of SSIT according to this review. The number of studies is limited, and the methodologies used are varied, making direct comparison difficult. In addition, they mostly report the effect on inactivation, but the mechanism of action remains unknown.

This review provides additional and updated information to the one published by Wozniak and Grinholc in 2018, and it is complementary to the review focused on in vivo studies published by our group in 2019 [23,24]. All three types of review agree on the promising approach of combining both therapies and the need to expand knowledge in this line.

The coating of surfaces such as catheters with antimicrobial drugs and aPDT are extremely effective and virtually overcome any resistance build-up. This is more complex on the skin and soft tissues because more variables become part of the process, especially with fungi. However, there is sufficient evidence to support this combined treatment strategy and to lay the foundation for this SSIT treatment approach [18].

Among the obstacles to the incorporation of aPDT as part of the SSIT-treatment, the need to require materials such as lamps for exciting the PS and the need to dedicate more time, because of the irradiation time and because often more than one session of aPDT is required are highlighted. However, it does not require much more clinician specialization, various studies have proven that lamps do not have to be especially expensive and specific, and sources of light with a wide irradiation spectrum or even daylight of radiation can be used effectively [18,23]. The methodology and evolution of these treatments need to be reported to the scientific community to continue expanding knowledge and increasingly implement this combination therapeutic option.

## 5. Conclusions

aPTD combined with antimicrobial agents is promising for the management of microorganisms that cause SSTI. It can help to fight them and to overcome AMR.

## Figures and Tables

**Figure 1 pharmaceuticals-14-00603-f001:**
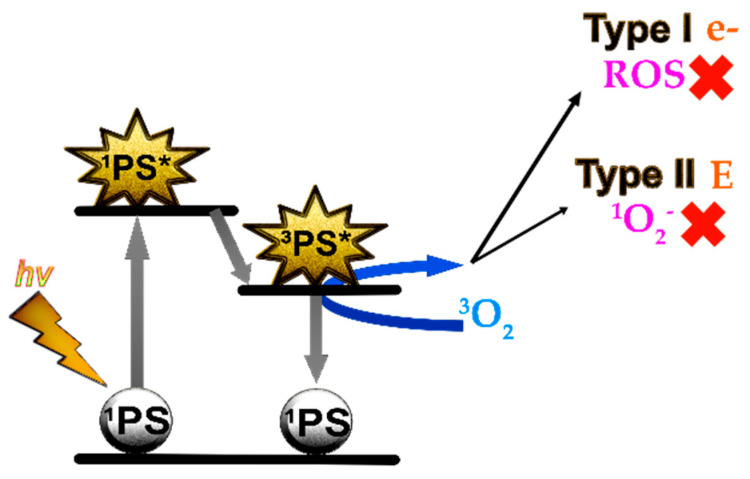
Mechanism of action of antimicrobial photodynamic therapy: photosensitizer molecules (PS) in ground state (^1^PS) are activated by light (*hv*) of a wavelength that matches the absorption wavelength of the molecule. PS reaches an excited state, first singlet (^1^PS*), very unstable, and rapidly triplet (^3^PS*). This reacts with the oxygen by transferring electrons (Type I) or energy (Type II) in its reversion to the ground state (^1^PS*). Type I reaction is characterized by the formation of reactive oxygen species (ROS) and Type II by the formation of singlet oxygen (^1^O_2_^−^). These molecules are highly reactive and are capable of damaging cells resulting in a toxic effect for microorganisms (Adapted with permission from ref. [18], 2018, Pérez-Laguna et al.).

**Table 1 pharmaceuticals-14-00603-t001:** Global priority list of antibiotic-resistant bacteria (Adapted from: https://www.who.int/medicines/publications/WHO-PPL-Short_Summary_25Feb-ET_NM_WHO.pdf?ua=1 (accessed on 20 June 2021)).

**Priority 1: CRITICAL**
-*Acinetobacter baumannii*, carbapenem-resistant -*Pseudomonas aeruginosa*, carbapenem-resistant -Enterobacteriaceae (*Klebsiella pneumonia*, *Escherichia coli*, *Enterobacter* spp., *Serratia* spp., *Proteus* spp., and *Providencia* spp., *Morganella* spp.), carbapenem-resistant, 3rd generation cephalosporin-resistant
**Priority 2: HIGH**
-*Enterococcus faecium*, vancomycin-resistant -*Staphylococcus aureus*, methicillin-resistant, vancomycin intermediate and resistant -*Helicobacter pylori*, clarithromycin-resistant -*Campylobacter*, fluoroquinolone-resistant -*Salmonella* spp., fluoroquinolone-resistant -*Neisseria gonorrhoeae*, 3rd generation cephalosporin-resistant, fluoroquinolone-resistant
**Priority 3: MEDIUM**
-*Streptococcus pneumoniae*, penicillin-non-susceptible -*Haemophilus influenzae*, ampicillin-resistant -*Shigella* spp., fluoroquinolone-resistant

**Table 2 pharmaceuticals-14-00603-t002:** Studies combining in vitro aPDT plus other treatment against *S. haemolyticus* and *S. epidermidis* comparing the methodology and the results biofilm studies, planktonic cell studies.

**Biofilm Ttudies**											
**Target**	**PS**	**PS Concentration (μM)**	**Antibiotic**	**Antibiotic Concentration (** **μ** **g/mL)**	**Source Type**	**Wavelength** **(nm)**	**Intensity (mw/cm^2^)**	**Fluence (J/cm^2^)**	**Inactivation Fraction (%)**	**Synergistic Observed Effect (*)**	**Reference**
*S. haemolyticus* clinical isolate	5-ALA	40	gentamicin	2	50-LED	630 ± 15	83	250	~70	>inactivation	[28]
*S. epidermidis* clinical isolate	5-ALA	40	gentamicin	2	50-LED	630 ± 15	83	250	~75	>inactivation	[28]
*S. epidermidis* RP62A & 5179R	TMP	10	vancomycin	200	tungsten lamp	400–800	166	300	~99.9999	>inactivation	[46]
**Planktonic Cell Studies**											
**Target**	**PS**	**PS Concentration (μM)**	**Antibiotic**	**Antibiotic Concentration (mg/L)**	**Preincubation Time (h)**	**Irradiation Time (h)**	**Source Type**	**Wavelength**	**Media**	**CFU/200 μL Well**	**Log_10_ Reduction**
*S. epidermidis* ATCC 35984	TAPP	5	chloramphenicol	2	19	5	100 W, 120 V Sylvania white light	Broad spectrum	TSB	~10^6^	~2
*S. epidermidis* ATCC 35984	TAPP	5	tobramycin	4.5	19	5	100 W, 120 V Sylvania white light	Broad spectrum	TSB	~10^6^	~3

(*): the combination causes an increase in percentage of microbial inactivation greater than the sum of the antibacterial activity of the aPDT plus the antibiotic treatment when they are applied alone; 5-ALA: 5-aminolevulinic acid; aPDT: antimicrobial photodynamic therapy; CFU: colony forming unit; LED: light-emitting diode; PS: photosensitizer; TAPP: meso-tetra (4-aminophenyl) porphine; TMP: tetra-substituted N-methyl-pyridyl-porphine; TSB: trypticase soy broth.

**Table 3 pharmaceuticals-14-00603-t003:** Comparison of methodology and results of an in vitro study of aPDT followed by culture with antibiotics on *Mycobacterium fortuitum*.

Strain	PS	PS Concentration (mg/mL)	Antibiotic	Antibiotic Concentration (mg/mL)	Preincubation Time (h)	Source Type	Wavelength (nm)	Intensity (mw/cm^2^)	Fluence (J/cm^2^)	Media/Culture	CFU/200μL Well	Synergistic Observed Effect (*)	Reference
*M. fortuitum* clinical isolate	MB	50	amikacin	0–0.5	0 + 72 with antib	metal halogen lamp	560–780	100	100	PBS with 0.02% Tween 80 / Muller Hilton	~10^8^	>inactivation (≥2 Log_10_ reduction)	[49]
			ciprofloxacin hydrochloride	0–0.06									
			moxifloxacin hydrochloride	0–0.06									

(*): the combination causes an increase in percentage of microbial inactivation greater than the sum of the antibacterial activity of the aPDT plus the antibiotic treatment when they are applied alone; aPDT: antimicrobial photodynamic therapy; MB: methylene blue; PBS: phosphate-buffered solution.

**Table 4 pharmaceuticals-14-00603-t004:** Studies on combination of in vitro aPDT plus other treatment against Gram-negative bacteria that cause SSTIs comparing the methodology and the results.

Strain	PS	Antibiotic	Phase	Source Type	Wavelength (nm)	Intensity (mw/cm^2^)	Fluence (J/cm^2^)	Synergy	Observed Effect (*)	Reference
*E. coli* ATCC 25922	ATAZTMPo	gentamicin	planktonic	LED (Sorisa Photocare)	638 ± 9 nm	17	45	yes	>inactivation	[27]
*E. coli* ATCC 25922	TAPP	tobramycin or chloramphenicol	planktonic	100 W, 120 V Sylvania white light	Broad spectrum	ND	ND	no	additivity	[32]
*E. coli* ATCC 25922	MB	ciprofloxacin	planktonic	IrradLED^®^ biopdi, São Carlos, SP, Brazil	~660	ND	2.8 and 5.6	yes	>inactivation	[35]
*E. coli* ATCC 25922	MB	ciprofloxacin	biofilm	IrradLED^®^ biopdi, São Carlos, SP, Brazil	660	ND	11.2 and 22	yes	>inactivation	[35]
*E. coli* ATCC 9027 and MDR clinical isolates	endogenous porphyrins	ciprofloxacin or norfloxacin n	planktonic	LED Dermaled^®^	~470 and ~625	ND	ND	yes	incrase in halo	[34]
*E. coli*	Chlorin e6	colistin, ciprofloxacin or amikacin	planktonic	diode laser, Laser Coupler 635 (Wroclaw, Poland)	635	290	120	yes	>inactivation	[51]
*E. coli*	Tetra-Py^+^-Me	ampicillin or chloramphenicol	planktonic	white light lamps (13 lamps OSRAM 21 of 18 W each)	Broad spectrum 380 to 700	40	-	yes	>inactivation	[52]
*E. coli, E. aerogenes*, and *K. pneumoniae* resistant to 3^rd^-cephalosporins, clinical isolates	MB	ceftriaxone	planktonic	LED (Biopdi/Irrad-Led 660)	660n ± 5	25	25	no	indifference	[53]
*P. aeruginosa* PAO1	TMP	tobramycin	biofilm	mercury vapor lamp	Broad spectrum	-	220–240	yes	>inactivation & tobramycin MIC decreased	[54]
*P. aeruginosa* PAO1	endogenous porphyrins	gentamicin	planktonic	Nd:YAG laser continuous / Pulsed-Q switched	532	106		yes	>inactivation	[55]
*P. aeruginosa* PAO1and others MDR and XDR	endogenous porphyrins	gentamicin, meropenem or ceftazidime	planktonic	Single-emitter diode lamp	410	15.7	50	yes	>inactivation & antibiotic MIC decreased	[56]
ATCC 27853 *P. aeruginosa* ATCC 27853	MB	ofloxacin	planktonic	LED	~637	44		yes	>inactivation	[57]
*P. aeruginosa* ATCC 27853	MB	gentamicin	planktonic	LED lamp (Showtec LED Par 64 Short 18 x RGB 3-in-1 LED, Highlite International B.V. Spain)	625 ± 10	7	18	yes	bactericidal effect with lower MB-PDT dose	[38]
P. aeruginosa ATCC 27853	MB	gentamicin	biofilms	LED lamp (Showtec LED Par 64 Short 18 x RGB 3-in-1 LED, Highlite International B.V. Spain)	625 ± 10	7	18	yes	bactericidal effect with lower MB-PDT dose	[38]
*P. aeruginosa* ATCC 10145 and 35 clinical isolates including MDR and XDR	RB	camel or pexiganan	planktonic	LED lamps (SecureMedia, Poland)	~514	23	60	yes	>inactivation	[58]
*P. aeruginosa ATCC 25668 and* sensitive and resistant clinical isolates	RB	sulfanilamide	planktonic	18 W white luminescent lamp	Broad spectrum 400–700	1.25	-	no	indifference	[26]
*A. baumannii* 2 XDR clinical isolates	RB	gentamycin, doxycicline, trimethoprim-sulfamethoxazole, ciprofloxacin, imipenem, piperacillin-tazobactam, ceftazidime, ampicillin-sulbactam, colistin	planktonic	LED	515	70	300	yes	>inactivation & antibiotic MIC decreased	[59]
*A. baumannii* 2 XDR clinical isolates	endogenous porphyrins	gentamycin, doxycicline, trimethoprim-sulfamethoxazole, ciprofloxacin, imipenem, piperacillin-tazobactam, ceftazidime, ampicillin-sulbactam, colistin	planktonic	LED	411	130	109.1	yes	>inactivation & antibiotic MIC decreased	[59]

(*) > inactivation: the combination causes an increase in the percentage of microbial inactivation greater than the sum of the antibacterial activity of the aPDT plus the antibiotic treatment when they are applied alone; additivity: the combination causes an increase in percentage of microbial inactivation equal to the sum of the antibacterial activity of the aPDT plus the antibiotic treatment when they are applied alone: Indifference was defined as no change from the most active antibiotic treatment; aPDT: antimicrobial photodynamic therapy; ATAZTMPo; 2-aminothiazolo[4,5-c]-2,7,12,17-tetrakis(methoxyethyl)porphycene; RB: rose bengal; LED: light-emitting diode; MB: methylene blue; MDR: multidrug resistant; MIC: minimum inhibitory concentration; ND: no data; PS: photosensitizer; TAPP: meso-tetra (4-aminophenyl) porphine; TBO: toluidine blue O; Tetra-Py^+^-Me: 5,10,15,20-tetrakis(1-methylpyridinium-4-yl)porphyrin tetra-iodide; TMP: meso-tetra (*N*-methyl-4-pyridyl) porphine tetra tosylate; XDR: extensively-drug resistant.

**Table 5 pharmaceuticals-14-00603-t005:** In vitro antimicrobial photodynamic therapy plus other treatment studies on infectious microorganism of skin and soft tissues grouped by photosensitizing family.

***S. aureus***						
**PS Group**	**PS**	**Antibiotic**	**Phase**	**Synergy**	**Observed Effect (*)**	**Reference**
Porphycene and Porphyrin	ATAZTMPo	gentamicin	planktonic	yes	>inactivation	[27]
	5-ALA	gentamicin	biofilm	yes	>inactivation	[28]
	DP	gentamicin, vancomycin, rifampin, fusidic acid	planktonic	no	additivity	[29]
	DP	oxacillin	planktonic	yes	oxacillin MIC decreased	[29]
	5-ALA	netilmicin, cefaclor, vancomycin	biofilm	yes	>inactivation	[30]
	TMP	vancomycin	biofilm	yes	>inactivation	[31]
	TAPP	vancomycin, ceftriaxone	planktonic	no	indifference	[32]
	TAPP	chloramphenicol, tobramycin	planktonic	yes-no	>inactivation MRSA; additive MSSA	[32]
	Tetra-Py+-Me	ampicillin	planktonic	yes	faster bactericidal effect	[33]
	Tetra-Py+-Me	ampicillin	pork skin (ex vivo)	yes	>inactivation	[33]
	(endogenous)	ciprofloxacin, norfloxacin	planktonic	yes	>inactivation	[34]
Phenothiazines	MB	chloramphenicol	planktonic	no	additivity	[32]
	MB	ciprofloxacin	planktonic	yes	>inactivation	[35]
	MB	ciprofloxacin	biofilm	yes	>inactivation	[35]
	TBO	linezolid	biofilm	yes	>inactivation	[35]
	MB	linezolid	biofilm	no	indifference	[35]
	TBO	gentamicin	planktonic	yes	>inactivation	[37]
	MB	gentamicin	planktonic	yes	bactericidal effect with lower MB-PDT dose	[38]
	MB	gentamicin	biofilm	no	no significant > inactivation	[38]
	MB	linezolid, mupirocin	planktonic	yes	bactericidal effect with lower MB-PDT dose	[39]
Xanthenes	RB	linezolid, mupirocin	planktonic	yes	bactericidal effect with lower RB-PDT dose	[39]
	RB	gentamicin	planktonic	yes	bactericidal effect with lower RB-PDT dose	[41]
	RB	gentamicin	biofilm	yes	>inactivation	[41]
	RB	methicillin	planktonic	yes	methicillin MIC decreased	[26]
***S. haemolyticus***						
**PS Group**	**PS**	**Antibiotic**	**Phase**	**Synergy**	**Observed Effect (*)**	**Reference**
Porphycene and Porphyrin	5-ALA	gentamicin	biofilm	yes	>inactivation	[28]
***S. epidermidis***						
**PS group**	**PS**	**Antibiotic**	**Phase**	**Synergy**	**Observed effect (*)**	**Reference**
Porphycene and Porphyrin	5-ALA	gentamicin	biofilm	yes	>inactivation	[28]
	TMP	vancomycin	biofilm	yes	>inactivation	[31]
	TAPP	chloramphenicol, tobramycin	planktonic	yes	>inactivation	[32]
***M. fortuitum***						
**PS Group**	**PS**	**Antibiotic**	**Phase**	**Synergy**	**Observed Effect (*)**	**Reference**
Phenothiazines	MB	ciprofloxacin, moxifloxacin or amikacin	planktonic	yes	>inactivation	[49]
***E. coli***						
**PS Group**	**PS**	**Antibiotic**	**Phase**	**Synergy**	**Observed Effect (*)**	**Reference**
Porphycene andPorphyrin	ATAZTMPo	gentamicin	planktonic	yes	>inactivation	[27]
	TAPP	tobramycin or chloramphenicol	planktonic	no	additivity	[32]
	endogenous porphyrins	ciprofloxacin or norfloxacin n	planktonic	yes	incrase in halo	[34]
	Tetra-Py^+^-Me	ampicillin or chloramphenicol	planktonic	yes	>inactivation	[52]
Phenothiazines	MB	ciprofloxacin	planktonic	yes	>inactivation	[35]
	MB	ciprofloxacin	biofilm	yes	>inactivation	[35]
	MB	ceftriaxone	planktonic	no	indifference	[52]
Chlorophylls	Chlorin e6	colistin, ciprofloxacin or amikacin	planktonic	yes	>inactivation	[51]
***P. aeruginosa***						
**PS Group**	**PS**	**Antibiotic**	**Phase**	**Synergy**	**Observed Effect (*)**	**Reference**
Porphycene and Porphyrin	TMP	tobramycin	biofilm	yes	>inactivation & tobramycin MIC decreased	[54]
	endogenous porphyrins	gentamicin	planktonic	yes	>inactivation	[55]
	endogenous porphyrins	gentamicin, meropenem or ceftazidime	planktonic	yes	>inactivation & antibiotic MIC decreased	[56]
Phenothiazines	MB	ofloxacin	planktonic	yes	>inactivation	[57]
	MB	gentamicin	planktonic	yes	bactericidal effect with lower MB-PDT dose	[38]
	MB	gentamicin	biofilm	yes	bactericidal effect with lower MB-PDT dose	[38]
Xanthenes	RB	camel or pexiganan	planktonic	yes	>inactivation	[58]
	RB	sulfanilamide	planktonic	no	indifference	[26]
***A. baumannii***						
**PS Group**	**PS**	**Antibiotic**	**Phase**	**Synergy**	**Observed Effect (*)**	**Reference**
Xanthenes	RB	gentamycin, doxycicline, trimethoprim-sulfamethoxazole, ciprofloxacin, imipenem, piperacillin-tazobactam, ceftazidime, ampicillin-sulbactam, colistin	planktonic	yes	>inactivation & antibiotic MIC decreased	[59]
Porphycene and Porphyrin	endogenous porphyrins	gentamycin, doxycicline, trimethoprim-sulfamethoxazole, ciprofloxacin, imipenem, piperacillin-tazobactam, ceftazidime, ampicillin-sulbactam, colistin	planktonic	yes	>inactivation & antibiotic MIC decreased	[59]
***Candida* spp.**						
**PS Group**	**PS**	**Antibiotic**	**Phase**	**Synergy**	**Observed Effect (*)**	**Reference**
Phenothiazines	MB	fluconazole	planktonic	yes/no	≥inactivation	[61]
	MB	miconazole	planktonic	yes	>inactivation	[62]
Porphycene and Porphyrin	TMP	miconazole	planktonic	yes	>inactivation	[62]
	TMP	fluconazole	planktonic	no	indifference	[62]
**Dermatophytes and moulds**						
**PS Group**	**PS**	**Antibiotic**	**Phase**	**Synergy**	**Observed Effect (*)**	**Reference**
Xanthenes	RB	clotrimazole	Planktonic (in vitro: spores)	yes	≥inactivation	[63]
Porphycene and Porphyrin	5-ALA	itraconazole	planktonic	yes	≥inactivation	[64]
Phenothiazines	MB	itraconazole, voriconazole, posaconazole, amphotericin	planktonic and biofilms	yes	MIC decreased	[15]
	MB	itraconazole	planktonic	yes	>inactivation	[69]

(*) > inactivation: the combination causes an increase in the percentage of microbial inactivation, greater than the sum of the antibacterial activity of the aPDT plus the antibiotic treatment when they are applied alone. Additivity: the combination causes an increase in the percentage of microbial inactivation equal to the sum of the antibacterial activity of the aPDT plus the antibiotic treatment when they are applied alone. Indifference was defined as no change from the most active treatment. 5-ALA = 5-aminolevulinic acid; ATAZTMPo = 2-aminothiazolo[4,5-c]-2,7,12,17-tetrakis(methoxyethyl)porphycene; DP = deuteroporphyrin; MB = methylene blue; MIC = minimum inhibitory concentration; MRSA = methicillin resistant *S. aureus*; MSSA = methicillin sensitive *S. aureus*; RB = rose bengal; PS = photosensitizer; TAPP = 5,10,15,20-tetrakis [4-(3-*N*,*N*-dimethylaminopropoxy) phenyl] porphyrin; TBO = toluidine blue O; TAPP = meso-tetra (4-aminophenyl) porphine; TMP = tetra-substituted *N*-methyl-pyridyl-porphine; Tetra-Py^+^-Me = 5,10,15,20-tetrakis (1-methylpyridinium-4-yl) porphyrin tetra-iodide.

**Table 6 pharmaceuticals-14-00603-t006:** Studies on combination of in vitro aPDT plus other treatment against fungus that cause SSTIs comparing the methodology and the results.

Strain	PS	Antifungal	Phase	Source Type	Wavelength (nm)	Intensity (mw/cm^2^)	Fluence (J/cm^2^)	Synergy	Observed Effect (*)	Reference
fluconazole-resistant *C. albicans* and *C. glabratai*	MB	fluconazole	planktonic	InGaAlP LED	nd	200	-	yes	>inactivation	[61]
fluconazole-resistant *C. krusei*	MB	fluconazole	planktonic	InGaAlP LED	nd	200	-	no	indifference	[61]
*C. albicans* SC5314	TMP	miconazole	planktonic	broadband visible light (Sylvania GRO-LUX, 15 W, part no. F15T8/GRO)	575–700	4	1	yes	>inactivation	[62]
*C. albicans* SC5314	TMP	fluconazole	planktonic	broadband visible light (Sylvania GRO-LUX, 15 W, part no. F15T8/GRO)	575–700	4	1	no	indifference	[62]
*C. albicans* SC5314	MB	miconazole	planktonic	broadband visible light (Sylvania GRO-LUX, 15 W, part no. F15T8/GRO)	575–700	4	7.2	yes	>inactivation	[62]
*T. rubrum* clinical isolate	RB	clotrimazole	Planktonic (in vitro: spores)	LED	530	13.4	12	yes	≥inactivation	[63]
*F. monophora* clinical isolates	5-ALA	itraconazole	planktonic	Zeiss KL 2500 LED	635	36.8	10	yes	≥inactivation	[64]
*E. dermatitidis*, *F. solani*, *F. oxysporum* clinical isolates	MB	itraconazole, voriconazole, posaconazole, amphotericin	planktonic and biofilms	LED	635 ± 10	100	12-24	yes	MIC decreased	[15]
*S. globosa* 5 clinical isolates	MB	itraconazole	planktonic	LED	640 ± 10	22.2	40	yes	>inactivation	[69]

(*) > inactivation: the combination causes an increase in percentage of microbial inactivation greater than the sum of the antibacterial activity of the aPDT plus the antibiotic treatment when they are applied alone. = inactivation: the combination has no effect on the percentage of inactivation; 5-ALA: 5-aminolevulinic acid; aPDT: antimicrobial photodynamic therapy; nd: no data; RB: rose bengal; LED: light-emitting diode; MB: methylene blue; PS: photosensitizer; TMP: meso-tetra (*N*-methyl-4-pyridyl) porphine tetra tosylate.

**Table 7 pharmaceuticals-14-00603-t007:** The highlight combinations and the best treatment protocols supported by the existing evidence of in vitro studies on combined aPDT therapies against SSTI-causing agents.

✓ The combination of different antibiotics with aPDT in general improves the efficacy against in vitro Gram-positive bacteria.
✓ Different antibiotics combined with aPDT using porphyrins, phenothiaziniums or RB have synergistic effects in vitro against Gram-positive bacteria being the combination with protoporphyrin the most studied.
✓ The combination of aPDT with gentamicin has been extensively tested in both Gram-positive and negative bacteria, reporting positive effects in all cases.
✓ The combination of MB-aPDT with sublethal concentrations of antibiotics seems to be a good option against mycobacteria.
✓ MB-aPDT combined with ciprofloxacin is the best option in vitro against *E. coli.*
✓ Endogenous porphyrins or MB based-aPDT used in combination with antibiotics is a promising option against in vitro *P. aeruginosa* regardless of its antibiotic resistance pattern.
✓ The administration of aPDT before antifungals seems to enhance their in vitro antimicrobial effect, especially against yeast and dermatophytes.

aPDT: antimicrobial photodynamic therapy; RB: rose bengal; MB: methylene blue.

## Data Availability

No new data were created or analyzed in this study. Data sharing is not applicable to this article.

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
