# Peer review of "Photodynamic Therapy Combined with Antibiotics or Antifungals against Microorganisms That Cause Skin and Soft Tissue Infections: A Planktonic and Biofilm Approach to Overcome Resistances"

_pharmaceuticals, 2021, doi:10.3390/ph14070603_

Round 1

Reviewer 1 Report

Dear editor

I carefully read the manuscript Vanesa Pérez-Laguna et al. „Photodynamic therapy combined with antibiotics or antifungals against microorganisms that cause skin and soft tissue infections: a planktonic and biofilm approach to overcome resistances". My opinion about this work is a positive one. This is an interesting piece of work reviewing the approaches of combined  antimicrobial photodynamic therapy (aPDT) in addition of antibiotics or antifungals to attack bacteria and fungi in vitro (both planktonic or biofilm forms) focused on those microorganisms that cause infections in skin and soft tissues.

The science and topic of the submission is suited for the journal. The topic of the manuscript is of great scientific impact. The present manuscript is well written and has a clear structure. The literature within is up to date and well cited.

Nevertheless there are some weak points which are to be considered for the preparation of the final manuscript version. The manuscript is suitable for publication in Pharmaceutics. I recommend the publication after a minor revision.

  1. The authors describe the combination of aPdT and antimicrobial agents.Right at the beginning of the publication, the authors describe the relevance of resistant bacteria in therapy.However, since antifungal agents + aPdT are also described in mauscript, relevant resistant fungi should also be described here.
  2. A clear evaluation of the results would be important for the reader.How effective is the combination therapy and there are differences between individual application.We know from our own experience that the coating of surfaces such ascatheters with antimicrobial drugs and aPdT are extremely effective and virtually overcome any resistance build-up.This is msore complex on the skin, especially with fungi.Here the authors could evaluate the reviewed data in an even more differentiated manner.

Author Response

[Pharmaceuticals] Manuscript ID: pharmaceuticals-1242179 - Pérez-Laguna et al. Photodynamic therapy combined with antibiotics or antifungals against microorganisms that cause skin and soft tissue infections: a planktonic and biofilm approach to overcome resistances” - R1

Dear Editors,

The manuscript was sent back to us with some parts highlighted in yellow for having a “High duplication rate”. These parts of the manuscript have been rewritten without changing the content of the meaning or kept as they were when it has not been possible to change.

Fisrt part of section 2.2. Study selection, data collection process & characteristics.

3.1.1. Staphylococcus aureus Section, sentence in lines 223-227.

3.3. Escherichia coli Section, sentences in lines 358-362 and 370-373.

Final part of section 3.4. Pseudomonas aeruginosa, sentence in lines 425-429

3.7. Dermatophytes and moulds Section, sentences in lines 482-484 and 485-487.

Dear Reviewer 1,

We appreciate your positive feedback.

Regarding to the point 1:

1.- The authors describe the combination of aPdT and antimicrobial agents. Right at the beginning of the publication, the authors describe the relevance of resistant bacteria in therapy. However, since antifungal agents + aPdT are also described in mauscript, relevant resistant fungi should also be described here.

Lines 76-79 and Supplementary table 2 have been added in section 1.2. Antimicrobial resistance in skin and soft tissue infections causal agents.

Regarding to the point 2:

2.- A clear evaluation of the results would be important for the reader. How effective is the combination therapy and there are differences between individual application. We know from our own experience that the coating of surfaces such as catheters with antimicrobial drugs and aPdT are extremely effective and virtually overcome any resistance build-up. This is more complex on the skin, especially with fungi. Here the authors could evaluate the reviewed data in an even more differentiated manner.

Section 4. Summary of evidence & limitations, one sentence in the first paragraph (line 518-520) and one paragraph (lines 537-541) have been added to comment tis aspects.

Dear Reviewer 2,

Thanks for your comments.

Regarding to

“This reviewer would like to the authors to include more detailed discussion to the clinical utility of current state of research, and limitations of aPDT as it stand for further development as clinically relevant therapies. What are the road blocks for its applications and what could the field do to overcome these challenges. i.e. what is new on the horizon that could potentially meet these challenges?”

Section 4. Summary of evidence & limitations, the last paragraph (lines 542-550) has been added to comment tis aspect.

Regarding the others comments,

The last article search has been updated: June 1st, 2021.

The formatting of italicized words has been revised throughout the text.

The sentences listed that needed to be reworded for clarity and conciseness, has been changed.

The Graphic abstract has been slightly modified to make it easier to interpret.

A figure illustrating the ROS mechanism of action has been added: Supplementary figure 1

Special thanks for your positive comment on table 1. Its realization took a lot of effort and we honestly appreciate the recognition.

                            Kind regards,         

                                    Vanesa Perez-Laguna

Reviewer 2 Report

This manuscript reviews the in vitro activities of aPDT in combination with antibiotics or antifungals, focusing on microorganisms that cause SSTIs. The review is comprehensive and meets its stated aims of serving as a point of reference for subsequent new research. This reviewer would like to the authors to include more detailed discussion to the clinical utility of current state of research, and limitations of aPDT as it stand for further development as clinically relevant therapies. What are the road blocks for its applications and what could the field do to overcome these challenges. i.e. what is new on the horizon that could potentially meet these challenges? This will significantly increase the impact of the review. The paper is generally well written and the methodologies used was sound and appropriate. However, it should be updated with the latest research, if any, as the last search was carried out more than six months ago in September 2020.

Minor corrections are needed, such as consistency with italicizing bacterial species, and "in vitro" for example. Some sentences need to be reworded for clarity and conciseness, some of them are listed below. Please proof-read before resubmission.

Graphical abstract:

larger font sizes would improve readability

Instead of "the 1st part" it's shorter to just say aPDT and easier to understand as an abstract

Need to explain what the tick means – i.e. complete kill? Effective? Suggest removing the tick with a better graphical representation of your intended meaning.

Lines

32 could be replaced with are “…SSTI are defined…”

51 rephrase “stands out for being”

53 “a percentage of SSTI…” what is the actual statistic?

57 remove “it includes”

65 What are the “special characteristics”?

79 include a figure illustrating the ROS mechanism of action would be helpful

102 please elaborate what you meant by “best combination”. Best needs to be quantified and defined in the context of your review.

Table 1: really like this detailed table

3.1.1 the first small introduction on S. aureus needs to be expanded, currently too general and lacks substance.

Author Response

(The authors gave the same response as above.)
